# Interoperable Nanoparticle Sensor Capable of Strain and Vibration Measurement for Rotor Blade Monitoring

**DOI:** 10.3390/s21113648

**Published:** 2021-05-24

**Authors:** Soo-Hong Min, Ying-Jun Quan, Su-Young Park, Gil-Yong Lee, Sung-Hoon Ahn

**Affiliations:** 1Department of Mechanical Engineering, Seoul National University, Seoul 08826, Korea; msh7799@snu.ac.kr (S.-H.M.); swimpark@snu.ac.kr (S.-Y.P.); 2Institute of Advanced Machines and Design, Seoul National University, Seoul 08826, Korea; jeonyj710@snu.ac.kr; 3Department of Mechanical Engineering, Kumoh National Institute of Technology, Gyeongbuk, Gumi 39177, Korea; gylee@kumoh.ac.kr

**Keywords:** interoperable sensors, nanoparticle sensors, strain and vibration monitoring, strain gauge, drone motion monitoring

## Abstract

Recent advances in nanomaterials technology create the new possibility to fabricate high performance sensors. However, there has been limitations in terms of multivariate measurable and interoperable sensors. In this study, we fabricated an interoperable silver nanoparticle sensor fabricated by an aerodynamically focused nanomaterial (AFN) printing system which is a direct printing technique for inorganic nanomaterials onto a flexible substrate. The printed sensor exhibited the maximum measurable frequency of 850 Hz, and a gauge factor of 290.62. Using a fabricated sensor, we evaluated the sensing performance and demonstrated the measurement independency of strain and vibration sensing. Furthermore, using the proposed signal separation algorithm based on the Kalman filter, strain and vibration were each measured in real time. Finally, we applied the printed sensor to quadrotor condition monitoring to predict the motion of a quadrotor.

## 1. Introduction

A sensor is a key enabler for the process or condition monitoring and optimization for an overall machine [1,2,3]. Data obtained by sensors and real-time feedback to a mechanical system enable the direct benefits for the quality and performance of the end-product. Hence, previous research related to sensor technologies have focused on expanding the physical quantity that can be measured and improving the performance of the sensor itself [4,5,6,7,8].

Especially, there have been several efforts to fabricate highly sensitive strain sensors using metal nanoparticles (NPs) [9,10,11,12,13]. Using the current transport mechanism based on current tunneling between nanogaps between NPs, drastic contact resistance change was available according to physical quantity to be measured [14,15,16,17,18].

According to Appendix A and Appendix A (Appendix A), it has been confirmed that existing studies have limitations in both high sensitivity and a measurable range. Furthermore, the study was not only insufficient but also limited in terms of high performance for a multivariate measurable sensor [19,20]. In the case of strain sensors, several researches on sensors with high sensitivity or a wide measurable range have been reported while other physical quantities such as vibration and temperature cannot be measured simultaneously [21,22,23,24,25,26,27].

This research has focused on the development of an interoperable sensor capable of strain and vibration sensing without the decrease of each sensing performance. Using an aerodynamically focused nanomaterial (AFN) printing system which is a direct printing technique for inorganic nanomaterials onto a flexible substrate in low vacuum and room temperature conditions, a conductive pattern composed of porous NPs was fabricated in several tens of micrometers scale [28]. Then, the sensing performance of the fabricated sensor was evaluated and real-time measurement independency of strain and vibration using proposed signal processing technique was validated. Finally, real-time quadrotor motion prediction was conducted using the data from the printed sensor.

## 2. Materials and Methods

### 2.1. Materials

We used Kapton polyimide film as an adhesive substrate to attach on a certain level of free-form surface and silver nanoparticles (576, 832, <100-nm diameter, Sigma-Aldrich, Saint Louis, MO, USA) as printing materials. The printing process is occurred by mechanical movements of substrate driven by multi-axis stage (SGSP20, Sigma Koki, Japan) with a velocity of 0.2 mms^−1^ which is controlled by LabVIEW 2015 and NI USB 6009 modules (National Instrument, Austin, TX, USA). The AgNPs erupted from a nozzle with an inner diameter of 150 µm (Taeha Co., Korea). After a wiring and soldering process using silver paste (conductive paste, 735,825, Sigma-Aldrich, Saint Louis, MO, USA) at the tip of printed line pattern, UV curable adhesives (UV-3300, Skycares Co., Gimpo, Korea) are wrapped for electrical insulation and mechanical protection from external stimulation.

### 2.2. Sensor Direct Printing

Figure 1a,b shows a schematic diagram of the AFN printing system and its process, respectively. The AFN printing system includes a vacuum chamber, nozzle, and nanomaterials feeder to generate an aerosolized nanoparticle beam. Using successive repetition of excitation and purging of aerosolized nanoparticles, aerosolized nanomaterials are aerodynamically focused when they erupt from the AFN system and directly accumulated on to a substrate governed by drag force, Saffman’s lift force, and centrifugal force [25].

Since it does not require any post-process include chemical etching and heat treatment, it is an environmentally friendly process with a high degree of freedom in design and manufacturing. It is especially suitable for the fabrication of a microscale porous pattern which could be applied to highly sensitive sensor fabrication. Furthermore, patterns could be reconfigured or repaired according to needs, which defeat the limitations of conventional processes such as photolithography or laser ablation.

Aerodynamic focusing is administered by evacuation of compressed air and time-scaling of excitation, optimizing process parameters is important to fabricate patterns as desired. A high upstream pressure tends to drop off Saffman’s lift effect when focusing the nanoparticles inside the nozzle and reduces the relaxation time, thus augmenting the nanoparticle beam in downstream of the nozzle. Hence, an excitation time of 10 ms and purging time of 90 ms remained for the process while the source pressure and chamber pressure were maintained at 1000 and 400 Pa, respectively.

Figure 1c,d shows the optical microscope image and the scanning electron microscopy (SEM) image of the fabricated sensor obtained by charge-couple device (CCD) cameras (EO-0813C, Edmond Optics, USA) and field-emission scanning SEM (AURIGA60, Carl Zeiss AG, Germany), respectively. The printed sensor has a porous structure of NPs inside the triangular shape of the cross section of the strip, which enables highly sensitive properties of sensor due to the drastic variance of resistance occurred by mechanical detachment among NPs. In comparison to conventional metal foil strain gauge, it was demonstrated that several micrometers strain can be measured by an AFN printed sensor in a previous study [29,30].

According to the previous study, it was demonstrated that the packing ratio of the pattern, which is defined as the volume ratio of printed nanoparticles to the surface border of the pattern, could be controlled by the scan velocity of the AFN printing process [29,30]. By measuring the disparity between the mass of the substrate before and after AFN printing by a precise microbalance (Sartorius AG, Germany) and surface profile by laser confocal microscope (OLS 4100, Olympus, Japan), the packing ratio was calculated as shown in Figure 1f. The scan velocity dominantly influences on not only the geometry of the printed pattern but also the packing ratio. By differing the velocity of the scan from 2 to 80 µm/s, the packing ratio decreased from 65% to 40%.

Furthermore, the decrease of the packing ratio occurred with the increase of relative electrical resistance as shown in Figure 1g. Since the high packing ratio decreased the distance among nanoparticles, electrical resistance increased as the number of electrical floating nanoparticles increased in the printed pattern. Hence, the scan velocity was maintained as 80 µm/s during the AFN printing process for highly sensitive sensor fabrication.

## 3. Results and Discussion

### 3.1. Sensing Performance Evaluation

Figure 2a shows the schematic diagram of the sensing mechanism for a NP-based strain sensor to explain the highly sensitive properties of the fabricated sensor. A simplified electron-tunneling model has often been used to explore the resistive responses of the NPs based strain sensor, represents that the relative resistance change is due to mechanical detachment between NPs. Mechanical segregation between NPs enables the drastic variances of relative resistance.

We evaluated the strain sensing performance based on a standardized four-points bending method. Both ends of the sensor were fixed to the mechanical jig and the center of the sensor was translated by a motorized stage (SGSP 20–85, Sigma Koki, Japan) with the initial gap between both ends at 40 mm. The images during translation were captured by a CCD camera (UI-2240SE, IDS Imaging Development Systems, Germany). The resistive responses of the printed sensor and commercial strain gauge (FLA-2-11-1L, Tokyo Sokki Kenkyujo, Japan) were captured by a data acquisition board (NI-USB-6009, National Instruments, USA) to gauge and estimate the applied strain within the experimental setup. The plots in Figure 2b show the resistive response by the applied strain. The response of a sensor exhibited that the sensitivity was changeable dramatically by varying the scan velocity of AFN printing. In an instant, sensitivity of 1056 was exhibited at an applied strain with a narrow strain range [24,25].

To not only evaluate strain sensing performance but also vibration sensing performance, the sensing performance evaluation for vibration was also conducted. The printed sensor was directly attached to a vibration shaker (Vibration testing shaker, TIRA GmbH, Germany) and relative resistance change was measured during a continuous vibration with various vibration frequencies as shown in Figure 2c. Figure 2d shows the frequency analysis results by fast Fourier transform (FFT) according to induced frequency. The average and standard deviation values of spectrum were calculated using quality factor (Q-factor) at 70.7%. Hence, it was demonstrated that the printed sensor was capable of not only strain sensing but also vibration sensing.

In order to verify that the strain and vibration could be separated each other, a statistical verification was conducted. First, the sensor data was divided by strain, vibration, and the random noise component for further analysis as shown in Equation (1) where *X_t_* is sensor signal, *S_t_* is strain component, *V_t_* is vibration component, and *E_t_* is normal random noise component.
*X_t_* = *S_t_* + *V_t_* + *E_t_*(1)

As shown in Equation (2), the null hypothesis *H*_0_, which denotes that the resistance change of the sensor was only influenced by strain component, was set. Otherwise, the alternative hypothesis *H*_1_ assumed that the resistance change of the sensor was influenced by both independent strain and vibration component as shown in Equation (3).
*H*_0_: *X_t_* = *S_t_* + *E_t_*(2)
*H*_1_: *X_t_* = *S_t_* + *Acos*(2*π**f_c_t*) + *Bsin*(2*π**f_c_t*) + *E_t_*(3)

To test the probability of hypothesis, analysis of variance (ANOVA) based on F-distribution was conducted to determine whether the means of strain and vibration are different [31]. The F-distribution which is continuous probability distribution based on beta function could be simply explained by the variation between sample means by variations within the samples. In consideration of the statistical degree-of-freedom of the sensor data, the variable was defined to follow the F-distribution without scaling as shown in Equation (4), where DF denotes degree of freedom that can vary in an analysis without breaking any constraints and SS denotes sum of square of variation. The sum of square of variation at frequency with 0 Hz was subtracted to separate the strain effects of interoperable sensor.
(4)$$\frac{(\sum_{i}DF_{i}-DF_{t}-1)SS_{t}/DF_{t}}{\sum_{i}SS_{i}-I\left(0\right)-SS_{t}}\sim F(DF_{t},\underset{i}{\sum }DF_{i}-DF_{t}-1)$$

By calculating probability value (*p*-value) which is the probability of obtaining test results at least as extreme as the results actually observed during the test by assuming that the null hypothesis is correct, it was decided to whether reject or accept null hypothesis [32]. At a significance level of 0.05, the null hypothesis was rejected, and the alternative hypothesis was accepted, which means measurement independency of strain and vibration was validated as shown in Table 1.

Figure 2e shows the proposed overall flowchart of signal process technique for real-time separation of strain and vibration. First, the raw data of the sensor signal was filtered by the Kalman filter which is a widely used algorithm for a series of measurement observed over time, containing statistical noise and other inaccuracies [33,34]. We estimated a joint probability distribution over the variables for each time frame of the prediction steps. Then we updated estimates using a weighted average with more weight being given to estimates with higher certainty. The filtered signal was considered to the strain value and the value obtained by subtracting the filtered signal from the raw data was perceived as a vibration signal. The FFT was conducted for vibration signal to estimate the vibration frequency.

### 3.2. Interoperable Strain and Vibration Measurement

To apply the vibration and strain simultaneously to the sensor, the experimental setup using a rotor was configured as shown in Figure 3a. Since the blade was mechanically bended during rotation of the rotor, we could apply the vibration and strain to the sensor by attaching the sensor to the fan blade. We connected the fan blade (D200-26A, Hawco, UK) to the BLCD motor (DC 12V-36V 775, Mabuchi Motor Company, Japan) capable of rotation up to 1400 Hz by regulation of the voltage which was controlled by DC power supply (DP30-05A, TOYOTECH, Korea). Sensor was directly attached to the back side of the fan blade and the resistance of the sensor was measure using Source meter (KEITHLEY 2450, Keithley Instruments, USA).

Figure 3b shows the estimated resistance by linear interpolation with respect to various frequencies of 200, 400, 600, and 800 Hz. As discussed above, the resistance data was filtered using the Kalman filter to estimate the strain as shown by the solid line in Figure 3d. The estimated strain was compared to the actual strain which was measured by commercial strain gauge (FLA-2-11-1L, Tokyo Sokki Kenkyujo, Japan) as indicated by the dotted line in Figure 3d. The strain value measured by the printed sensor and measured by a commercial strain gauge exhibited similar trends, which denotes that the strain measurement performance was validated. The mechanical deformation of fan blade was initially increased and stagnated after few seconds varied by rotation frequency.

Figure 3d represents the estimated vibration signal by subtracting filtered data in Figure 3c The FFT results of the vibration signal exhibited that the peak frequency followed the induced frequency as shown in Figure 3e which denotes that vibration sensing performance of printed sensor was also validated.

### 3.3. Quadrotor Monitoring

We applied the interoperable strain and vibration sensing of the printed sensor to the commercial quadrotor (Tello, Ryze Technology, China). Figure 4a shows the schematic illustration of the drone and its motion including up and down, turn, and flip which were all supported by the manufacturer. The experiments were conducted by measuring the resistance during each motion and the spectrogram was calculated.

Figure 4b shows the photograph of experimental setup. We directly attached the printed sensors onto the backside of the blade similar to fan blade experiment. Similarly, a loop wire was hooked to the ring wire which is located in the shaft of the blade and the resistance of the shaft was measured. An identical source meter (KEITHLEY 2450, Keithley Instruments, USA) was used to measure the resistance of the sensor.

Figure 4c–e shows the spectrogram of time–frequency plots resulted by FFT of vibration signal of the sensor with respect to motion of up and down, turn, and flip, respectively. According to Figure 4c, the rotation frequency of the quadrotor was increased as the quadrotor went up and decreased during the quadrotor went down. For the turning motion, rotation frequency was the highest when it rotated. Hence, it has been confirmed that real-time monitoring was available for quadrotors including drones using the printed sensor. For the up motion of the drone, the frequency should increase to get more thrust for the elevation of the height. Otherwise, frequency should decrease to decrease their thrust force which was shown well in Figure 4c. In terms of the turn motion, four motors at each blade of the drone should be driven differently. With the angular speed of the motor in charge of the rotating axis being maintained, the angular speed of the motor in the direction to be rotated should increase and that of the other motors should decrease. In the experiment, the measured resistance of the blade we measured by the developed sensor was almost maintained during the turning motion since it behaved as a rotational axis. Finally, for the flipping motion, the angular speed of the motors driving the two blades in the direction to be turned upward should decrease and the angular speed of the other motors should increase. Likewise, the blade we measured during the experiment which was flipped in the direction of downward showed a decrease in motor frequency.

## 4. Conclusions

This research has focused on the fabrication and evaluation of an interoperable sensor which is capable of real-time strain and vibration measurement. The sensor was printed using an AFN printing process which is a dry and direct printing technique without chemical process using successive repetitions of excitation and purging of aerosol. The porous properties of the conductive patterns fabricated by the AFN printing process overcame the challenge of high sensitivity of current strain sensor and enabled the interoperable measurement of strain and vibration sensing.

The strain and vibration sensing performance of the printed sensor was evaluated respectively and independency was demonstrated by statistical methods. Furthermore, the signal processing technique for real-time separation was introduced based on the Kalman filter. As a result of experiments with a sensor attached to the rotating fan blade, the real-time interoperable measurement performance was verified. Finally, we applied the printed sensor to the quadrotor condition monitoring and predicted the quadrotor’s motion through the sensor data.

The proposed sensor exhibited the wide range in terms of gauge factor (GF) and maximum measurable frequency in comparison to the previous interoperable sensor as shown in Figure 5 [10,16,17,24,26,29,35,36,37,38,39,40,41,42]. The results presented in this research are expected to facilitate the highly sensitive reliable measurement of strain and vibration sensing based on a cost-effective method that can leverage the future development of related technologies including structural dynamic behavior analysis of rotational machines.

## Figures and Tables

**Figure 1 sensors-21-03648-f001:**
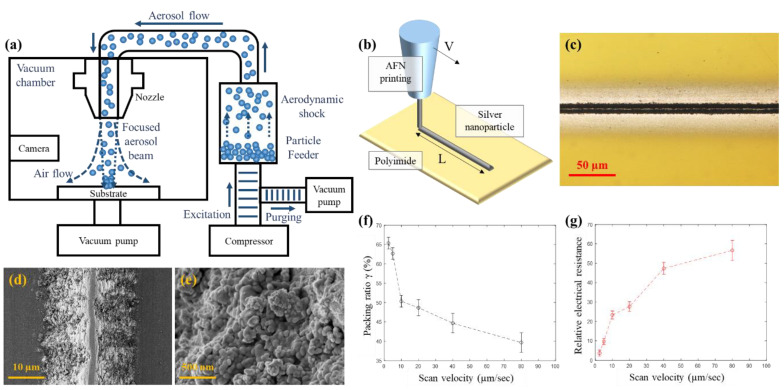
(**a**) Schematic diagram of the aerodynamically focused nanomaterial (AFN) printing system. (**b**) Schematic illustration of AFN printing of silver nanoparticle onto polyimide substrate. (**c**) Optical microscope image of AFN-printed microscale conductive line pattern. (**d**,**e**) Scanning electron microscope images of AFN-printed microscale conductive pattern. (**f**) Nanoparticle packing ratio with respect to the scan velocity. (**g**) Relative electrical resistance with respect to the scan velocity.

**Figure 2 sensors-21-03648-f002:**
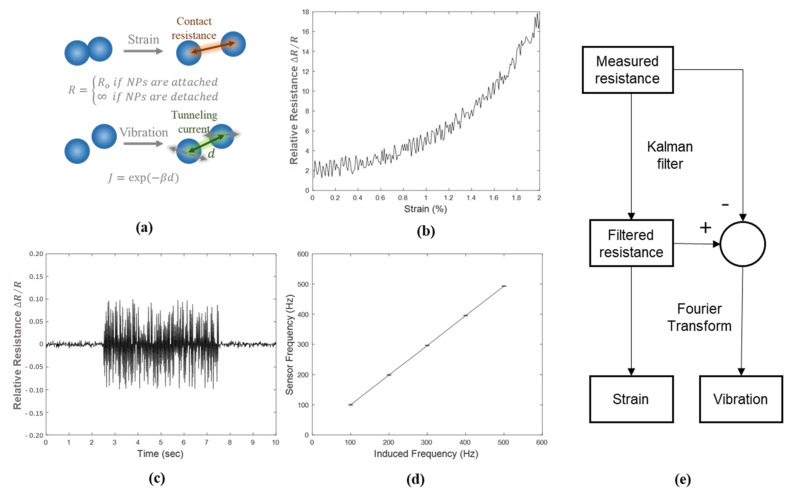
(**a**) Schematic diagram of the sensing mechanism of the nanoparticle sensor for strain and vibration measurement, respectively. (**b**) Relative resistance changes of sensor according to strain. (**c**) Relative resistance changes during continuous vibration with a frequency of 200 Hz by the vibration shaker. (**d**) Peak frequency of sensor resistance obtained by fast Fourier transform (FFT) at Q-level of 70.7% with respect to the induced frequency. (**e**) Flowchart of proposed interoperable strain and vibration measurement.

**Figure 3 sensors-21-03648-f003:**
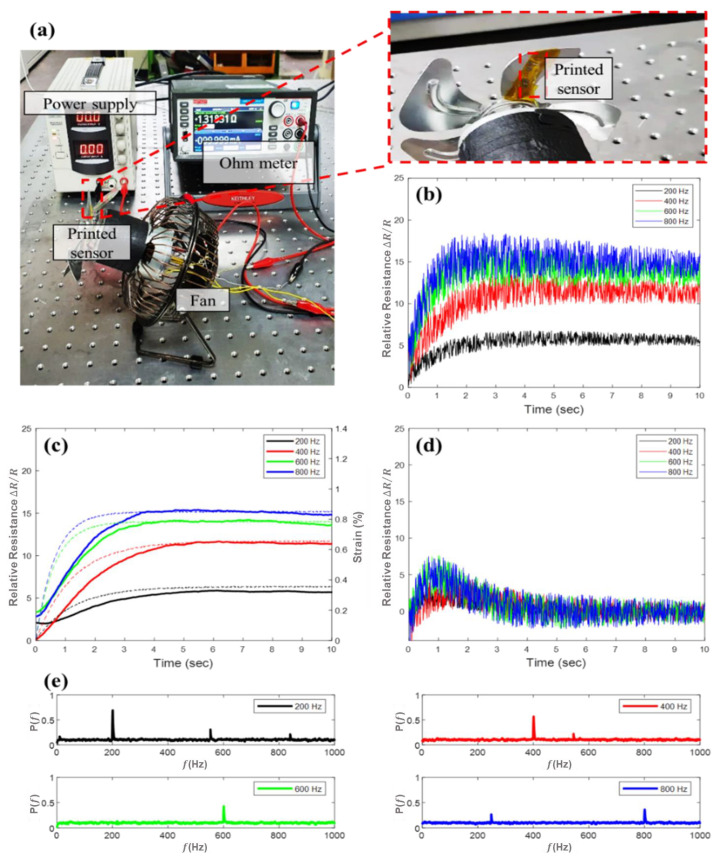
(**a**) Photograph of the experimental setup for strain and vibration measurement of the fan blade (**b**) Measured resistance according to rotor frequency. (**c**) Estimated strain from the printed sensor using the Kalman filter (solid line) and measured strain from the commercial strain gauge (dotted line) (**d**) Estimated vibration obtained by subtracting the estimated strain from the measured resistance. (**e**) Fast Fourier transform (FFT) results of estimated vibration.

**Figure 4 sensors-21-03648-f004:**
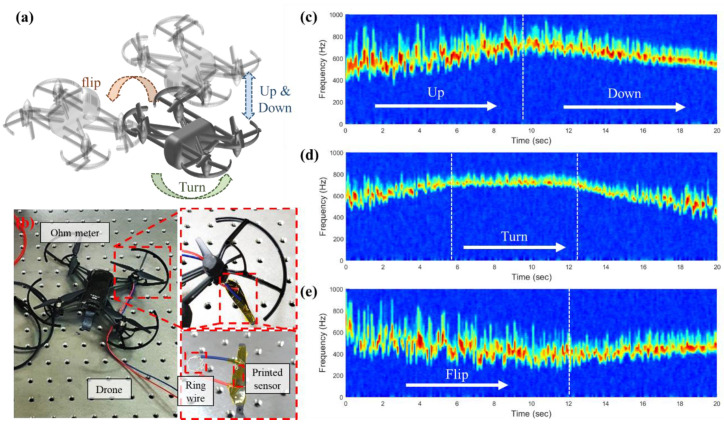
(**a**) Schematic illustration of drone and its motion including up and down, turn, and flip. (**b**) Photograph of experimental setup for rotor blade monitoring. Time-frequency plots of measured resistance with motion of (**c**) up and down, (**d**) turn, and (**e**) flip.

**Figure 5 sensors-21-03648-f005:**
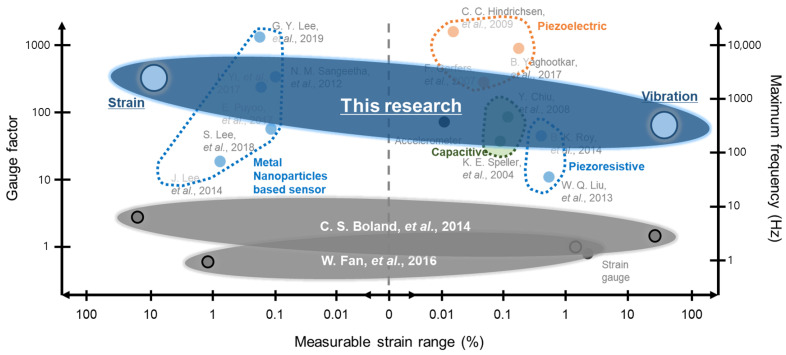
Comparison to previous studies according to sensitivity and maximum frequency.

**Table 1 sensors-21-03648-t001:** The results of *p*-value according to induced frequency.

Vibration Source (Hz)	Results of *p*-Value
200	0.032
400	0.018
600	0.011

## Data Availability

The data presented in this study are available in Appendix A here.

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
