# Peer review of "Interoperable Nanoparticle Sensor Capable of Strain and Vibration Measurement for Rotor Blade Monitoring"

_sensors, 2021, doi:10.3390/s21113648_

Round 1

Reviewer 1 Report

  • 4page, 129 line: What does F-distribution mean?
  • Page 4, 132 line: The definitions for D, S, Si, and I in Equation 4 are not explained.
  • Page 6, Figure 3(d): There is no distinction between the data measured with the Kalman filter and the commercial strain gauge.
  • Page 7, Figure 4(b): There is no explanation on the driving device for implementing the quadrotor's motion mode. And when I connect the printed sensor to the ring wire through a wire, I am curious if the wire does not affect the dynamic motion when it rotates.
  • Page 7, Figure 4(c-e): It is necessary to explain why the frequency behavior over time of the measured data in Figure 4(c-e) according to the motion mode of the quadrotor occurs. Also, due to the change in the speed of motion of the quadrotor and the disturbance such as wind, the time-frequency characteristics are expected to change even in the same motion condition. Please explain if you think it is possible to predict the motion.

Reviewer 2 Report

In this research, Min et al. introduced an interoperable nanoparticle (NP) sensor for strain and vibration measurements. By printing silver NP traces over polyimide substrate using the aerodynamically focused nanomaterials (AFN) method, a porous, conductive trace can be achieved for monitoring strain and vibration. Taking advantage of a signal separation algorithm, real-time strain and vibration measurements can be decoupled. As a demonstration, the authors used their printed sensor on a commercial quadrotor to differentiate and measure different modes of rotor blade motion.

In general, the research is well designed with considerable amount of novelty. I recommend acceptance after the authors consider the following comments/suggestions:

  1. The introduction contained some necessary background information, but I believe a more comprehensive literature review is required. Most importantly, the authors need to be careful when defining the following terms: "sensor technology", "multivariate measurable sensor", "high performance". I can easily think of tons groups who conduct excellent research in developing multivariate measurable sensors with high performance. Also, physical quantities such as vibration and temperature are not hard to integrate to strain sensors. I suggest the authors re-evaluate what is the actual contribution of this work in the realm of sensor technology, before abruptly jumping to statements such as "Despite the recent advances in sensor technologies, the study was not only insufficient but also limited in terms of high performance for multivariate measurable sensor".
  2. Section 2.1, why is Kapton polyimide film used as the substrate for the printed AgNPs? Is that the reason why the maximum measured strain of the sensor is only 2% (as shown in Fig. 2b)? If so, have the authors considered using more compliant and stretchable substrates for higher strain measurements?
  3. A follow up of 2, what is the failure mechanism for strain measurements at 2%? 
  4. Figure 2, how many sensors are tested in each experiment. Are the data shown from one single measurement? Currently no standard deviation is illustrated.
  5. Did the authors observe sensor drift? This can be more critical than other properties listed in this paper (sensitivity, gauge factor, etc.).

Reviewer 3 Report

General comments:

The manuscript presents the fabrication and characterization of a strain gauge made by silver nanoparticles. The authors obtained a high gauge factor of around 290 and they discuss the proposed method for extracting measurements of mechanical strain and vibration.

It is an interesting manuscript, but the “Materials and Methods” and “Results and discussion” sections are confusing in some parts, as noted below. In my opinion, the text needs careful revision to improve the understanding and value the results.

Specific comments:

The keywords are very generic. I suggest: Interoperable sensors; Nanoparticle sensors; Strain and Vibration monitoring; Strain gauge; Drone motion monitoring.

Lines 72 to 77: What “triangular shape” do the authors refer to? I think this is important because the authors assign the high sensitivity of the NP sensor to this feature. In addition, what is the “previous study” mentioned in lines 76 and 77?

Lines 107 to 110: How does Fig. 2b allow the sensitivity values of 18.60, 290.62 and 1056 to be obtained?

Fig. 2c should be better described in the legend. It does not show the resistance variation for various vibration frequencies, as stated in line 115.

Lines 120 to 139: The authors demonstrated statistically that the sensor signal can be divided into “strain + vibration + noise”. But I understood that the vibration experiment was done without strain load. Moreover, where are the measured signals for 200, 400 and 600 Hz? The reader needs to see graphically what the statistics indicate numerically.

Line 155: Is the motor rotation 1400Hz or 1400 rpm?

Fig. 3b is confused. It is better to remove the graph, because the text only mentions the dynamic measurement method.

Line 167: Are the frequencies 200, 400, 600, and 800 Hz induced by the vibration shaker? If so, how does it work together with the fan motor? If no, how do you induce these vibrations? Are these values the motor rotation? (800 Hz = 48000 rpm!).

Round 2

Reviewer 3 Report

In lines 72-73 I suggest: “The printed sensor has a porous structure of NPs inside the triangular shape of the cross section of the strip, which enables....”

Lines 107 to 110 must be rewritten because the sensitivity values of 18.60, 290.62, and 1056 cannot be obtained from Fig. 2b, as stated by the authors.

The authors replied that they had removed the confusing Figure 3b, but the revised manuscript shows the Figure 3 exactly the same as the first version of the manuscript.

In general, the authors partially made the corrections and improvements that I pointed out in the first revision. I consider that the manuscript still needs improvement for better understanding by readers.
